# Carbon Storage in Cropland Soils: Insights from Iowa, United States

Jim Jordahl [1,*], Marshall McDaniel [2], Bradley A. Miller [2], Michael Thompson [2], Sebastian Villarino [2,3] and Lisa A. Schulte [4]

1. Bioeconomy Institute, Iowa State University, Ames, IA 50011, USA
2. Department of Agronomy, Iowa State University, Ames, IA 50011, USA; marsh@iastate.edu (M.M.); millerba@iastate.edu (B.A.M.); mlthomps@iastate.edu (M.T.); sh4@iastate.edu (S.V.)
3. Consejo Nacional de Investigaciones Científicas y Técnicas (CONICET), Buenos Aires B1657, Argentina
4. Department of Natural Resource Ecology and Management, Iowa State University, Ames, IA 50011, USA; lschulte@iastate.edu
* Correspondence: jjordahl@iastate.edu

**Abstract:** The restoration of soil organic matter (SOM, as measured by soil organic carbon (SOC)) within the world's agricultural soils is imperative to sustaining crop production and restoring other ecosystem services. We compiled long-term studies on the effect of management practices on SOC from Iowa, USA—an agricultural region with relatively high-quality soil data—to highlight constraints on detecting changes in SOC and inform research needed to improve SOC measurement and management. We found that strip-tillage and no-tillage increased SOC by 0.25–0.43 Mg C ha$^{-1}$ yr$^{-1}$ compared to losses of 0.24 to 0.46 Mg C ha$^{-1}$ yr$^{-1}$ with more intensive tillage methods. The conversion of cropland to perennial grassland increased SOC by 0.21–0.74 Mg C ha$^{-1}$ yr$^{-1}$. However, diversifying crop rotations with extended rotations, and supplementing synthetic fertilizer with animal manure, had highly variable and inconsistent effects on SOC. The improved prediction of changes in SOC requires: the use of methods that can identify and disentangle multiple sources of variability; looking beyond total SOC and toward systematic collection of data on more responsive and functionally relevant fractions; whole-profile SOC monitoring; monitoring SOC in long-term studies on the effect of multiple conservation practices used in combination; and deeper collaboration between field soil scientists and modelers.

**Keywords:** soil organic carbon stock; soil management; SOC measurement; SOM; variability; scaling





## 1. Introduction

International efforts to limit global warming to 1.5 °C by 2100 are falling short of targets [1], leading to an increasing focus on $CO_2$ removal to meet climate goals [2,3]. Soil organic carbon (SOC) is an important component of the global carbon (C) balance, and increasing SOC is a form of $CO_2$ removal, though of limited durability (<100 years) [2]. Some argue that efforts to increase SOC for climate mitigation are misplaced given limitations on measuring and managing this pool [4], but the pace and impacts of anthropogenic climate change requires the consideration of all options. Furthermore, the low capacity and high price of more durable $CO_2$ removal technologies (e.g., direct air capture and C capture and storage) results in nearly all of the current estimated 2 Gt yr$^{-1}$ of $CO_2$ removal being due to land management [2].

SOC associated with croplands is highly relevant because of the global extent of agriculture, which covers between 10% and 15% of the total ice-free land area of the Earth, and because these lands are already highly managed. Furthermore, increased SOC can improve the resilience of croplands to extreme weather events associated with anthropogenic climate change [5]. Increasing SOC storage can meet both climate change mitigation and crop production goals, and is aligned with farmers' long-term interests. Yet,

farmers' ability to simultaneously manage for both crop production and SOC storage has largely been undervalued until recently [6]. The importance of measuring, managing, and monetizing SOC is coming into clearer focus with the development of numerous private sector C programs [7–9], corporate net-zero claims requiring documentation of supply chain (i.e., Scope 3) emission reductions [10], USDA initiatives such as the Climate-Smart Commodities Partnership program [11], and provisions of the Inflation Reduction Act [12], such as tax incentives (45Z) for biofuel producers to lower their C intensity, including C impacts of their feedstocks.

SOC is the dominant constituent of soil organic matter (SOM) and C typically accounts for more than half of SOM's dry mass [13]. SOM affects the physical, chemical, and biological properties and functions of soil [14–16]. SOM increases the capacity of soils to store and provide water and nutrients to support plant growth [17]; influences soil structure, soil erosion, water quality, and food security; and decreases the negative impacts of human activities to ecosystems [14]. Paradoxically, some of the key benefits of SOM, such as supplying nitrogen to plants, rely on its continued degradation rather than its accumulation [18–20]. This paradox highlights the ongoing challenge of optimizing soil management with respect to increasing SOM. Developing scientifically sound recommendations for managing these dynamic processes and pools requires long-term experiments and large amounts of high-quality data.

There are numerous challenges and opportunities in achieving, maintaining, and measuring increases in SOC to achieve climate goals and restore soil functions. Our objectives were to consolidate and summarize data from previously published studies on the baseline SOC stocks and the long-term effect of common conservation practices on carbon sequestration as SOC in Iowa, USA, including the ability to detect change and rates of change in SOC, and compare these results to the global literature on SOC response to agricultural management. A further goal was to summarize key issues and associated challenges and opportunities to achieve the reliable, measurable, and sustainable storage of carbon in these soils. Our hope is that knowledge generated from Iowa, a relatively data-rich portion of the world, can facilitate more informed SOC data collection strategies elsewhere.

## 2. Iowa Soils and Agriculture

The US state of Iowa is widely known for its highly productive, high-organic-matter soils (Table 1, Figure 1). This soil environment has been produced by a nearly ideal intersection of climate, soil parent material, and native vegetation. The state has a humid continental climate with hot summers and cold winters. Mean annual precipitation ranges from 590 to 900 mm, and mean annual temperatures range from 6 °C to 10 °C. Much of the geologic parent material from which the soil was formed has a high silt content (median = 61%; [21]), resulting in soil textures that are silt loam, silty clay loam, or an adjacent texture class. A loamy soil, especially with silt's ideal particle size, provides abundant water-holding capacity at pressure potentials that are readily available for plant uptake. The silty and fine-textured particle size distributions of Iowa soils may also be partly responsible for large SOM stocks, as finer particle sizes promote more persistent SOC [22,23]. Before the 19th century, the native vegetation across much of the state was tallgrass prairie, dominated by diverse and deep-rooted grasses and forbs.

**Table 1.** Descriptive statistics for silt content (%), soil organic matter (%), and bulk density (g cm$^{-3}$) in Iowa from 0 to 30 cm as mapped in the USA's Soil Survey [21]. SOC stock was calculated by assuming a 2:1 ratio of SOM and SOC. Although SOC is known to exist below the topsoil (A horizon), this estimation of SOC stock focuses on the thickness of soil with visibly evident organic carbon accumulation as indicated by "mollic" colors.

| Measure | Min | Max | Mean | STD | Median | 90th Percentile |
|---|---|---|---|---|---|---|
| Silt (%) | 1 | 90 | 54 | 15 | 61 | 71 |
| Sand (%) | 0 | 96 | 19 | 17 | 9.5 | 41 |
| Clay (%) | 0 | 60 | 27 | 6 | 28 | 34 |
| SOM (%) | 0 | 85 | 4 | 3 | 3 | 6 |
| Bulk Density (g/cm$^3$) | 0.2 | 1.7 | 1.3 | 0.1 | 1.3 | 1.5 |
| Mollic Thickness (cm) | 0 | 203 | 45 | 36 | 43 | 94 |
| SOC Stock (Mg/ha) | 0 | 2309 | 120 | 132 | 95 | 294 |

Note: Silt, sand, clay, SOM, and bulk density are based on 0–30 cm data. Mollic thickness and SOC stock are variable, with stock being variable with thickness.

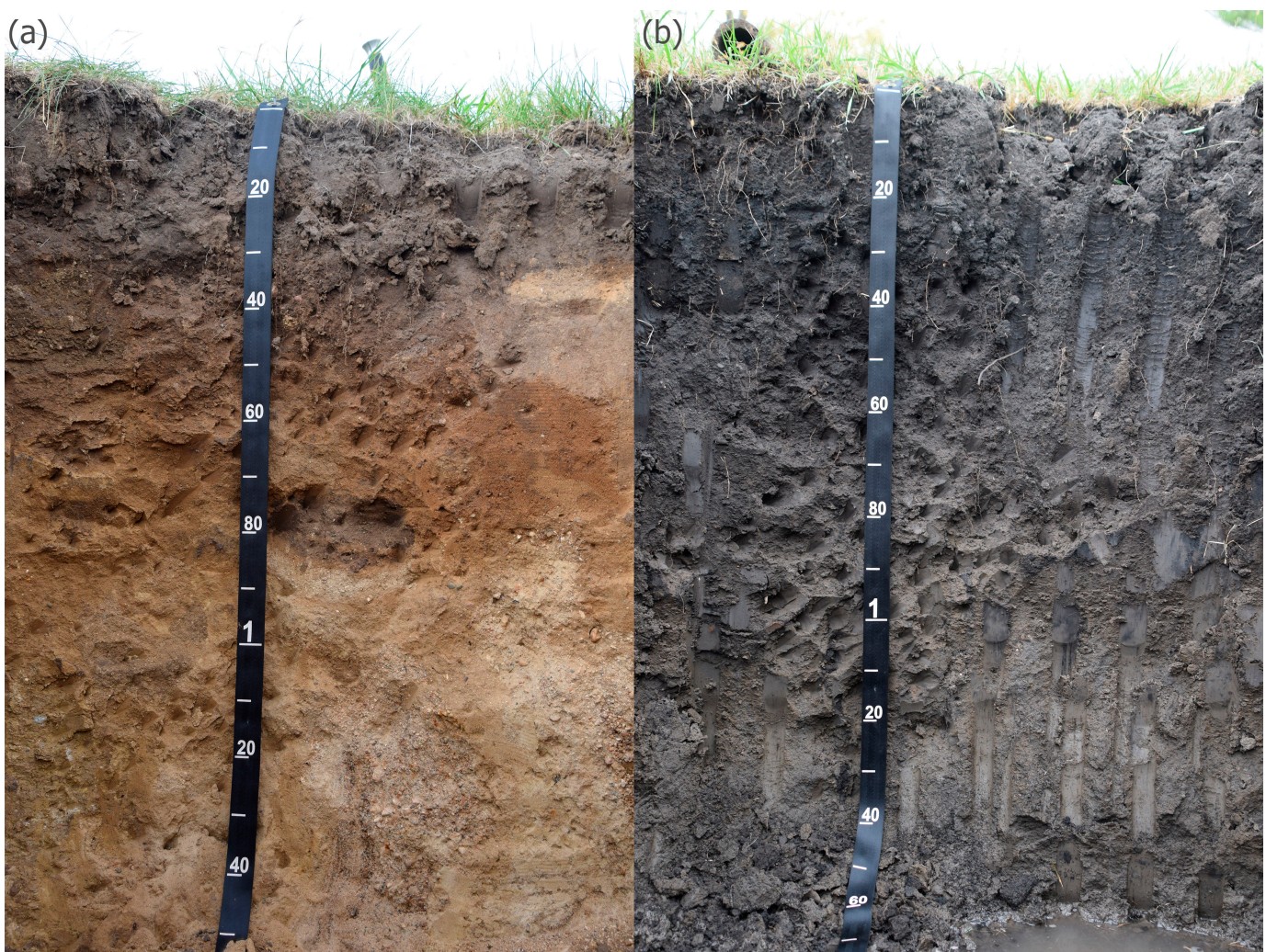

**Figure 1.** Variable thickness of mollic colors for two common soil series from the intensively drained Des Moines Lobe Landform in north-central Iowa. The (**a**) 30 cm mollic colors profile is a Clarion (fine-loamy, mixed, superactive, mesic Typic Hapludoll), and the (**b**) 60 cm mollic colors profile is a Webster (fine-loamy, mixed, superactive, mesic Typic Endoaquoll). Photos by BA Miller, Iowa State University.

Information about Iowa's soils is a result of over a century of investment from county, state, and federal levels of government. Although many more soil observations were made as part of this cooperative effort, a database of over 8000 deep (>1 m) soil cores documents this work [24]. By combining field experience with aerial photography and then field verification, soil scientists produced soil survey maps at cartographic scales of 1:15,840 and 1:12,000 [25]. These soil maps are among the most detailed in the world, especially for covering such a large extent. The maps were produced county by county (~2330 km$^2$), with teams of four soil scientists dedicated to surveying each county over a four-year period (Fenton, personal communication).

Iowa's soil and climate conditions are conducive to high yields of summer annual crops. The landscape is dominated by the cultivation of maize and soybeans, used for livestock feeds, vegetable oils, sweeteners, and biofuel feedstock [26,27]. Humans have lived in North America since as early as 5000 BCE, practicing various forms of agriculture [28–30]. Since Euro-American settlement in the 19th century, however, Iowa's topsoil has been drastically altered by the conversion to field crop production. A substantial portion of topsoil has been redistributed by wind, water, and tillage erosion. Even on gently rolling landscapes such as those of north central Iowa, the loss of organic-matter-rich topsoil from some landscape positions is evident [31]. One study based on remotely sensed data has suggested that 37% of Iowa's agricultural fields have no remaining topsoil [32], but this does not account for local sediment movement and deposition. Considering the thickness of topsoil in prairie remnants, spatial modeling suggests that the SOC stock in Iowa soils before Euro-American settlement was about 1.47 billion Mg [9]. A similar analysis calculated from the gSSURGO database [21] suggests the present stock is about 990 million Mg (Figure 2), which indicates a loss of about 32%. Losses and gains are a function of landscape position, with the largest losses measured in the backslope [31]. Preliminary data also suggest a loss of C from upland swales and depressions. Despite slow but continued increases in the adoption of soil conservation practices by farmers [33], the estimated annual statewide average soil loss is 13 Mg ha$^{-1}$ yr$^{-1}$ [34].

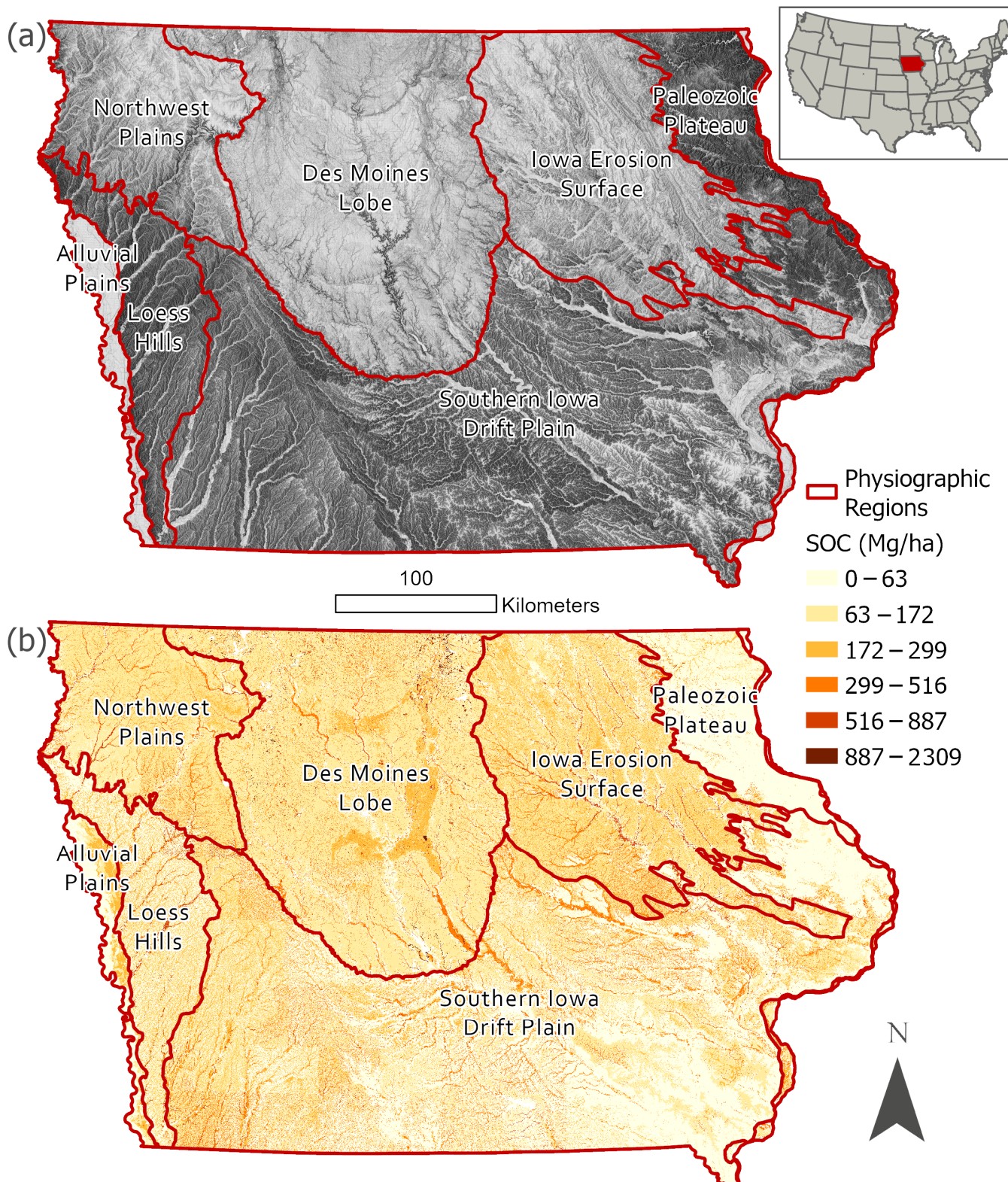

**Figure 2.** (**a**) The relief of Iowa, USA, displayed by a hillshade model derived from LiDAR elevation data (State of Iowa, 2009), which heavily influences the distribution of (**b**) soil organic carbon (SOC). Estimates of SOC content is based on gSSURGO data [21]. The depth included is variable and is based on the thickness of soil with visible evidence of organic carbon accumulation (i.e., "mollic" colors, a standard for indicating soil most enriched by organic matter). For some soil areas, this thickness may be up to 2 m.

## 3. Agricultural Management Practices and Soil Organic Carbon: Long-Term Studies in Iowa

On a worldwide basis, SOC stocks have declined by 133 Pg C since the beginnings of cultivation, with the "Corn Belt", of which Iowa is the center, among the areas of greatest loss [35]. While we cannot point to one cause, this SOC loss can be attributed to soil-tillage-induced erosion, extensive tile drainage, improper soil fertility, and increased soil temperatures [36,37]. A long-term study on SOC loss from 23 counties in Iowa reported a net loss of SOC from 0 to 100 cm depth (−10.8%) but gain at 100 to 150 cm (of +1.5%) between the years of 1959 and 2007 [31]. This study represents the status of most Iowa farms regarding SOC, and we do not expect this rate of loss from the upper soil profile to have slowed since the 2007 sampling.

As in other locations around the world, Iowa researchers and farmers have looked toward multiple practices to increase SOC. These include (but are not limited to): reduced tillage [38], residue management [39], diversified crop rotations [40,41], manure or other organic carbon amendments [38,42], using winter cover crops [43], growing perennial crops [44,45], using a living mulch or perennial groundcover [46], and full or partial land conversion to perennial vegetation [47]. These practices reliably reduce soil erosion and improve water quality, but long-term data from these studies suggest that consistent measurement of absolute changes in SOC remains challenging. Here, we summarize patterns in extensive, relatively long-term SOC datasets associated with three practices and collected in Iowa (Table 2).

**Table 2.** Summary of approaches and methods used in long-term studies of three practices intended to increase soil organic carbon (SOC) in Iowa, USA.

| Practice Intended to Increase SOC | Number of Sites/Studies | Length of Studies' SOC Monitoring (years) | Sampling Depth (cm) | Approach | Treatments | References |
|---|---|---|---|---|---|---|
| Reduced Tillage | 7 | 12 | 0 to 60 | Sampled annually, with baseline * | 1. Moldboard Plow<br>2. Chisel Plow<br>3. Deep Rip<br>4. Strip Tillage<br>5. No Tillage | [48] |
| Diversified Crop Rotations | 3 | 10–12 | 0 to 30–90 | Sporadic sampling, no baseline * | 1. Maize-Soybean<br>2. Maize-(Maize)-Soy-Oat-Alfalfa | [40,41,49–53] |
| Conversion to Perennial Vegetation | 4 | 7–40 | 0 to 15–30 | Various, chronosequence, most without baseline * | 1. Cropland<br>2. Restored Prairie<br>3. Native (or Remnant) Prairie | [45,47,54,55] |

* Baseline refers to sampling before, or shortly, after practice change. Without baseline sampling, determining net SOC loss/gains is more challenging (see ref. [56]).

### 3.1. Reduced Tillage

Although the use of no-till in Iowa has increased significantly since the 1980s, full-width tillage is still used on the majority of cropland in Iowa for at least part of the crop rotation [57,58]. Intensive tillage transiently decreases soil bulk density and increases soil aeration, mixes crop residues with mineral soil, and breaks soil aggregates and crop residues into smaller pieces—all of which increase decomposition (and loss of SOC as $CO_2$). Tillage also dries and warms soils more rapidly in springtime by increasing aeration and decreasing albedo. These soil microclimate benefits and increased decomposition of residues are why many growers in Iowa continue to till—converting to no-tillage has been shown to have a 7.6% maize yield penalty according to a global meta-analysis [59]. However, the SOC benefits to reducing tillage in Iowa are compelling [48] (Figure 3).

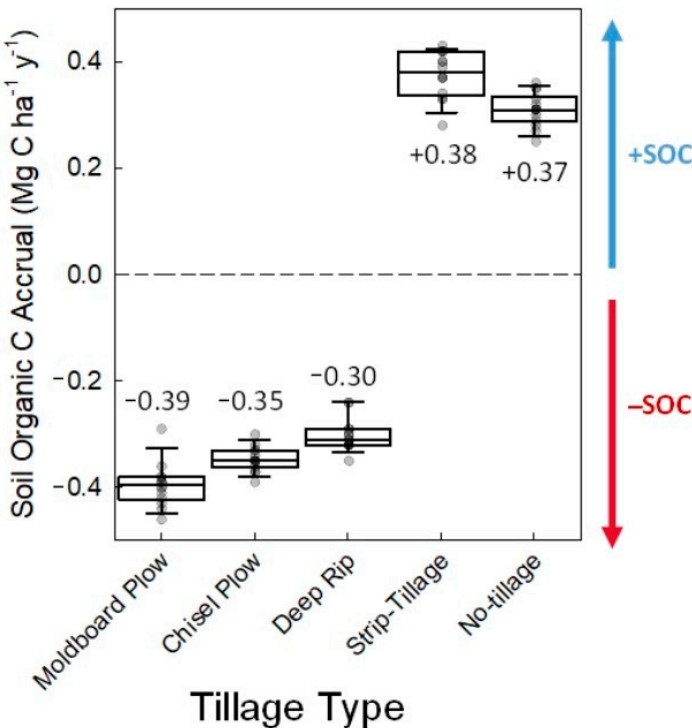

**Figure 3.** Five tillage types and impact on soil organic carbon (SOC) at 0–60 cm depth from 2002 to 2014. Boxplots include seven sites spread throughout Iowa and two rotations (maize-soybean and maize-maize-soybean) for a total of *n* = 14. Mean state-wide averages are shown with the boxplots. Data from [48].

Al-Kaisi and Kwah-Mensah (2020) [48] compared five tillage methods—moldboard plow, chisel plow, deep rip, strip-tillage, and no-tillage, over a 12-year period across seven locations in Iowa (Figure 3). Soils were sampled and analyzed for SOC in increments to a depth of 60 cm. Prior to the initiation of the study, all sites had been in row crop production using common conservation tillage methods for at least 50 years.

Averaged across seven locations and two rotations, no-tillage and strip-tillage systems added 0.25 to 0.43 Mg SOC $ha^{-1}$ $y^{-1}$ to a depth of 60 cm. Conversely, chisel plow, deep rip, and moldboard plow had losses ranging from −0.24 to −0.46 Mg SOC $ha^{-1}$ $y^{-1}$ over the 12-year study (Figure 3). SOC gains and losses predominantly occurred in the upper 30 cm. Interestingly, there was no tillage interaction with crop rotation or location, despite varying climates and soils [48]. Taken together, this comprehensive Iowa study indicates that reducing tillage improves SOC storage, with a reasonable expectation that, if no-tillage and strip tillage systems replaced more intensive tillage systems across Iowa, SOC stocks in the upper 60 cm of soil could increase at an average net rate of approximately 0.7 Mg SOC $ha^{-1}$ $yr^{-1}$ ([average SOC increase with strip-tillage or no-tillage] + [average avoided SOC loss caused by more intensive tillage systems]).

Globally, the majority of studies have also consistently documented the positive impact of reducing or eliminating tillage on SOC stocks, at least in the upper 30 cm. This is probably best illustrated via meta-analyses, which document a range from <0.1 to 0.57 Mg C $ha^{-1}$ $y^{-1}$ increase in SOC [60–63]. For Iowa climate and soil conditions, Ogle et al. [64] reported a 0.33 Mg C $ha^{-1}$ $y^{-1}$ increase in SOC in the 0–60 cm soil depth following conversion to no-till. While there is some scientific debate, [65–68], most data syntheses point to net SOC accrual (e.g., [63,69,70]). In addition to soil depth, the other many relevant factors affecting SOC response to adopting no-till include climate, soil type, cropping system, and duration [60,61,63,64,69,71]. The potentially nonlinear temporal response of SOC to no-till conversion also poses particular challenges to the accurate measurement and interpretation of results [60].

In addition to increasing SOC, reduced tillage provides multiple environmental co-benefits. Several studies have demonstrated reduced erosion from no-tillage or reduced tillage [72–74]. Soil infiltration [75], structure and porosity [69,76], and microbial biomass and activity [77] also improve. Conversion to no-till also reduced nitrate-nitrogen leaching by 14 ± 10% at a site in northwestern Iowa [78].

### 3.2. Diversified Rotations

Diverse crop rotations that include small grains and forage crops were widely used across much of Iowa until the 1950s, after which intensive monocultures of continuous maize and maize-soybean became increasingly dominant [79]. A concern with this increased specialization in crop production is the potential negative effects on SOC and other aspects of soil health. Three long-term crop rotation experiments examined the effects (Table 2).

These three experiments compare a conventional maize-soybean rotation with a diversified maize-soybean-oat-alfalfa rotation. The Kanawha experiment was established in 1954 in northern Iowa (42°94′ N, 93°17′ W); the Nashua experiment was established in 1979 in northeast Iowa (42°95′ N, 92°54′ W); and the Marsden experiment was established in 2002 in central Iowa (42°01′ N, 93°47′ W). Detailed information on site history, management, and soil sampling for SOC can be found in several publications [40,41,50,80]. Tillage and fertilization vary somewhat for the three sites, but full-width tillage was used at all sites. Animal manure was incorporated after the alfalfa phase of the rotation at the Marsden site to simulate a diversified, integrated crop-livestock system [80].

Unlike reduced and no-tillage, there are no clear trends in the impacts of crop diversification on multiple soil measures. The coefficients of variation for bulk density, SOC concentration, and SOC stock are high (medians range from 5 to 8%) for both maize-soybean and diversified rotations (Figure 4). Although yield data and allometric equations suggest that below-ground inputs of carbon were 20–35% greater in the diversified rotations compared to the maize-soybean rotation, SOC increases within the total soil profile were minor [41]. Although changes in SOC stock were not consistently observed, changes in certain C fractions were noted. For example, C stored in particulate organic matter (POM), microbial biomass C, and salt-extractable organic carbon were significantly increased with diverse rotations [52,53]. One reason for the lack of a clear trend in SOC may be fertilization: Russell et al. [50] found that nitrogen fertilization tends to negate the effect of increased below-ground inputs of carbon in diverse rotations (SOC increases are maximized at the agronomic optimum nitrogen application rate in Iowa intensive maize and maize-soybean rotations [81]). Furthermore, the formation of mineral-associated organic matter (MAOM) requires substantial amounts of nitrogen, which could pose another limitation for its accumulation [82].

Meta-analyses report SOC increases for both diversified crop rotations [83] and manure inputs [84]. The lack of SOC response to diversified rotations in Iowa is puzzling but similar to recent findings from Wisconsin, a neighboring US state, where the researchers found even 29 years of diversified management was not enough to detect SOC changes between conventional cropping and diversified crop rotation [85]. Climate or soil conditions may make SOC slow to respond to cropping system diversification in this region, if there is a response at all.

Even though changes in SOC stock are difficult to discern, there are a plethora of additional benefits of diversified cropping systems. The four-year diversified rotation in the Marsden experiment dramatically lowered synthetic nitrogen inputs (−86%), herbicide use (−50%), and energy use (−65%); improved many metrics of soil health including reduced soil erosion (−62%); increased crop yields (maize +5%, soybean +20%); and even improved air quality [52,53,86,87].

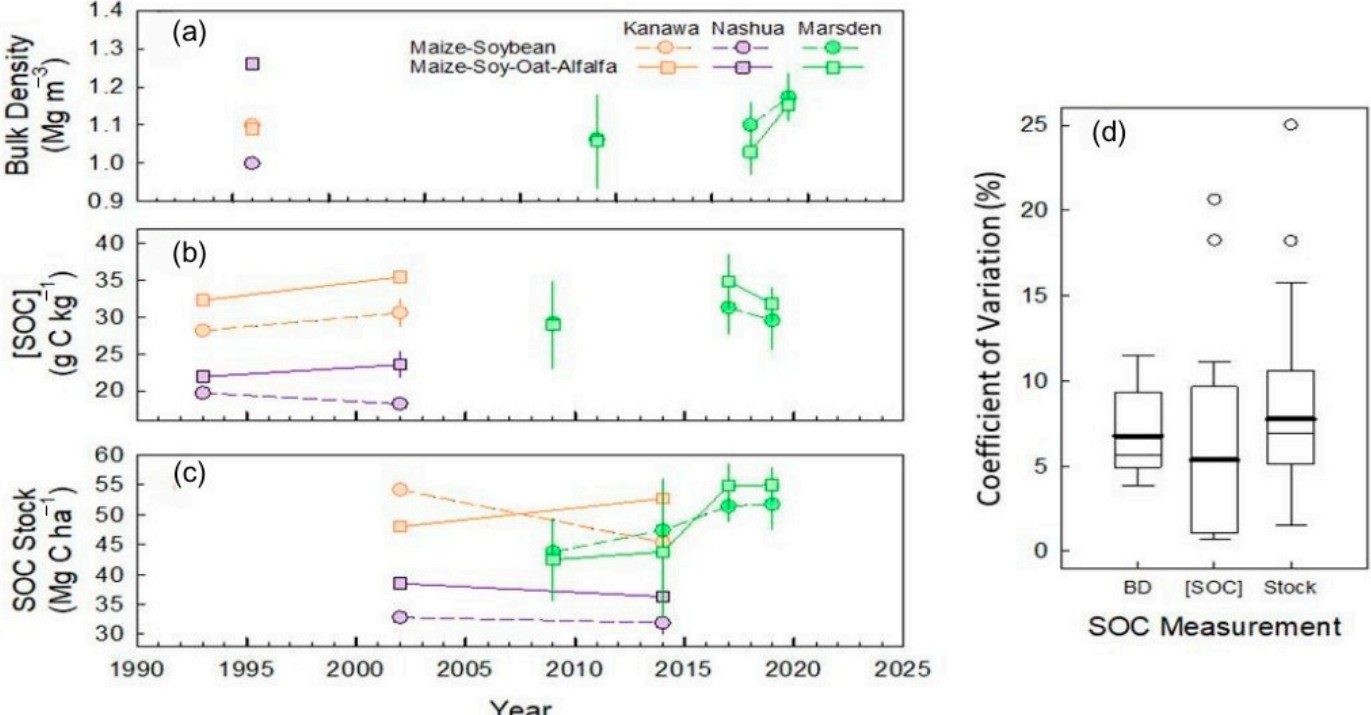

**Figure 4.** (**a**) soil bulk density, (**b**) soil organic carbon (SOC) concentration, (**c**) calculated SOC stock, and (**d**) the coefficient of variation for all three soil measurements at 0–15 cm depth across time from three long-term crop diversification experiments. Means and standard deviations are compiled from [40,41,49–53], and the coefficient of variation was calculated by treatment and year (*n* = 6 for BD, *n* = 10 for [SOC], and *n* = 12 for SOC stock). The thick dark lines in boxplots are the means.

### 3.3. Converting Cropland to Perennial Vegetation

Perennial grasslands, called "prairie," once covered 70–80% of Iowa's landscape, but now comprise < 1% [88]. Remnant prairies average 30–60% greater SOC than adjacent cultivated cropland [55,89], suggesting a massive potential sink for SOC with conversion of cropland to native prairie. Numerous environmental benefits have been shown to accrue with the restoration of even modest areas of prairie within Iowa's cropland landscapes, including increased biodiversity, reduced runoff, and increased retention of nitrogen and phosphorus [90]. Converting cropland to perennial grassland vegetation may be particularly beneficial on soils that are marginally productive for annual crops [91], and can be economically feasible if farmers and/or farmland owners receive federal or state cost-share to assist with establishment and management costs, and enroll in a private sector C program [92].

Long-term studies of the effect of conversion of cropland to perennial grassland on SOC in Iowa (Table 3) have included three approaches: (i) a chronosequence approach, sometimes called space-for-time substitution because it involves sampling different fields (ideally under very similar soils and climates) that have been converted to perennial vegetation [47,54,55], (ii) randomized complete block designs that ideally measured SOC at baseline or beginning of the conversion and some years later [45], and (iii) sampling through long time frames (>50 years) and monitoring SOC loss [31].

**Table 3.** Summary of four studies conducted in Iowa, USA on the effect of cropland conversion to perennial grassland on soil organic carbon stocks and change (at 0–15 cm).

| Study Citation | Region in Iowa (Nearest City) | Soil Texture | Classification Based on USDA's Soil Taxonomy | Years since Conversion | Cropland SOC (Mg C ha$^{-1}$) | Restored Perennial Grassland SOC (Mg C ha$^{-1}$) | Native Prairie Remnant SOC (Mg C ha$^{-1}$) | SOC Accrual Rate (Mg C ha$^{-1}$ y$^{-1}$) | Significant Difference between Crop and Restored Grassland |
|---|---|---|---|---|---|---|---|---|---|
| [47] | Central (Prairie City, IA) | Silty clay loam | Aquertic and Typic Argiudolls | 14 | 44 | 35–39 | 55 | 0.74 | Variable |
| [54] | Central (Prairie City, IA) | Silty clay loam | Oxyaquic, Aquertic, and Typic Argiudolls | 21 | 25 | 40 | 58 | 0.59 | NA |
| [45] | Central (Boone, IA) | Clay loam | Typic Endoaquoll, Aquic Hapludoll | 7 | 55 | 60 | NA | 0.71 [†] | No |
| [55] | North central (Various) | Various | Various | 40 | 49 | 54 | 60 | 0.21 | Yes |

[†]: Prairie biomass is harvested. Also had fertilized prairie that increased at 0.29 Mg C ha$^{-1}$ on average (not significantly different at $p < 0.05$). Larger differences between years than between treatments.

The SOC accrual rate in these four long-term studies ranged from 0.21 to 0.74 Mg C ha$^{-1}$ y$^{-1}$, but SOC in the grasslands was not always significantly greater than in cropland (Table 3). Even after multiple decades (up to 40 years), SOC did not reach levels of nearby undisturbed prairies [47,54,55]. These findings highlight the slow recovery of SOC, and the patience needed to observe "statistically significant" differences between the annual croplands and re-established grasslands. Also, it is important to note that in none of the studies did the perennial grassland reach the SOC stocks measured in the remnant prairie benchmark (Table 3).

The reason for slow and variable recovery of Iowa's SOC from conversion to perennial vegetation, sometimes resembling something like native vegetation, is unclear (Table 3). Variability in soil properties, like clay concentration, limited the ability to detect significant differences in switchgrass plantations compared to other covers [93]. Tile drainage of somewhat poor, and poorly drained sites may promote residue decomposition by aerating the soils for longer periods than under native conditions. Warm-season grass residues may decompose faster than those of cool-season species [92]. Ten-month laboratory incubation studies of central Iowa soils have shown that root residues from unfertilized reconstructed prairie plots dominated by warm-season grasses decomposed rapidly and did not lead to carbon storage in the soil [94]. Reducing soil erosion and increasing the duration of living plant roots and the quantity of other root-derived C inputs are likely the main factors that increase SOC under perennial vegetation [95]. Furthermore, the speed at which the SOC concentrations increases when tilled soils are converted to grassland can be dependent on hillslope position [55]. The more eroded, shoulder-slope positions tended to accrue SOC faster than positions lower down the hillslope [55], perhaps because the change in SOC concentration was easier to discern from the lower background concentration on the eroded sites. Changes in inorganic carbon are often not considered, but impacts can be significant. Conversion from cropland to grassland reduced soil inorganic C by 14.4% [96].

Global meta-analysis has indicated that the conversion of cropland to pasture resulted in increased SOC stocks, especially in the upper soil horizons, but the increases do not reach that of the native conditions on the timescales of most scientific studies [97]. Kristersen et al. [98] note the potential role of incorporating large herbivores in establishing increased and resilient stocks of SOC in grasslands globally. Research interest in understanding potential applications in Iowa is increasing, but data on net GHG impacts and SOC changes in grazed systems in Iowa are scant [9]. Overall, Fargione et al. (2018) [99] estimate modest potential contributions from grassland restoration to climate mitigation as compared to avoided grassland conversion.

## 4. Implications for Soil Organic Carbon Measurement and Management

The results of these studies suggest that, in Iowa soils, SOC responds slowly to management and has a high variability, such that changes in SOC are difficult to detect with statistical certainty. Here, we discuss the implications of these findings for SOC measurement and management within our broader understanding of soil science literature. We focus on SOC measurement, spatial and temporal variation in SOC, and other considerations.

### 4.1. What to Measure and Why

Detecting changes in SOC is difficult. Even long-term, well-controlled studies reflect this challenge for some management practices, as we show here for diversified rotations and conversion from cropland to perennial grassland. Several factors influence our ability to detect and interpret SOC changes based on soil management, including the depth measured, bulk density, and the influence of different SOC fractions including highly persistent fractions (naturally occurring or added "biochar").

#### 4.1.1. Soil Depth

The majority of Iowa research studies on the effect of management practices on SOC have focused on the upper 0–15 or 0–30 cm, leaving substantial uncertainty on impacts to deeper layers and changes in whole profile stocks [9]. Of the long-term studies summarized here, only two assessed SOC deeper in the soil profile. Al-Kaisi et al.'s [48] measured to a depth of 60 cm and established a positive influence of no-tillage on SOC across seven study locations. Poffenbarger et al. [41] did not find a consistent effect of cropping system on the vertical distribution of SOC within the soil profiles among three long term trials. More studies including measurements of deeper soil layers may help establish more consistent patterns, especially patterns of SOC that might be stable over long periods of time. Soil carbon sampling protocols and programs are generally focused on SOC changes in the upper 30 cm [8], but we note that subsurface horizons warrant considerable additional investigation because stocks of subsoil C may be substantial in some soils. Those deeper stocks may still be influenced by soil management at the soil surface, and the biochemical and physical processes that stabilize C in subsoils proceed at different rates from those in topsoils [100].

#### 4.1.2. Bulk Density

Bulk density (dry mass per volume) is an essential parameter for the calculation of changes in SOC stocks, but it is difficult and time-consuming to obtain accurate measurements. Uncertainties associated with bulk density measurements are related to both spatial variability and the variety of methods that can be used. These uncertainties can easily exceed those associated with SOC measurement [9]. Automated in-situ tools such as multi-sensing penetrometers that predict bulk density rapidly and to multiple depths have been developed (e.g., [101]), but their use in soils with coarse fragments, gravel, or rocks is uncertain. There has been little testing of these tools in Iowa soils.

#### 4.1.3. SOM Fractions

The components of SOM differ in both chemical and physical characteristics, and their functions and behaviors in soils also differ. Therefore, soil scientists have long investigated ways to chemically, physically, biologically, or through a combination of the three approaches, fractionate total SOM to better understand soil C and explain its behavior. Separating SOM according to its size (and sometimes density) into particulate organic matter C (POM-C) and mineral-associated organic matter C (MAOM-C) has proven useful for this purpose, given that these fractions broadly differ in their persistence, formation pathway, and soil function [102–104]. The POM fraction represents organic particles, usually partially decomposed fragments of plant tissue, that are >50–63 μm in diameter (depending on the specific protocol that is followed). MAOM is empirically defined as organic matter that is finer than 53–63 μm. POM may be located freely between soil

particles or occluded in large aggregates [104]. MAOM comprises organic compounds that may be chemically bound with soil minerals or could be located within micropores or small aggregates (<50–63 μm). The degree of its close physical association with silt and clay particles in an undisturbed soil is likely to be highly variable and dependent on clay type and concentration, pH, metal oxides, and exchangeable cations, as well as faunal activity that intimately mixes organic and inorganic components. In most fractionation schemes, MAOM includes microbial biomass and necromass. Soil organisms—both fauna and microorganisms—that decompose organic residues and environmental conditions regulate the formation and decay of these fractions. Understanding the complex interactions between these SOM pools, the environment, and decomposers is key to choosing the best management strategies to optimize SOM storage [105].

In Iowa croplands, the majority of SOC (>90%) is stored as MAOM-C [41]. Some of the organic matter in this fraction can persist in the soil for long periods (e.g., decades or longer) due to its attachment to or occlusion by minerals, but some MAOM compounds are also readily accessible to microbially derived enzymes and are decomposable. Some soil minerals with a low specific surface area can become saturated with MAOM-C [106–108]. To the extent that MAOM is composed of microbial biomass and necromass, its formation depends on the availability of nitrogen, which could pose another limitation for its accumulation [82]. When the mineral phase of a soil becomes saturated with C-containing compounds, additional C inputs can be stored as POM [107]. Thus, in such situations, it may be recommended to focus on managing POM (e.g., by cultivating crops that add low-quality residues to the soil) [105]. However, determining the C saturation limit for particular soils remains challenging and is subject to ongoing debate [109].

On the other hand, labile fractions like POM have been suggested as useful indicators of soil health compared to total SOM, because they exhibit greater sensitivity to changes in land use and management [110]. Results from a three-year study of different crop systems across a topographic gradient in Iowa indicated that while total SOC stocks remained unchanged, POM-C increased [111]. Additionally, soil aggregation showed a positive correlation with POM stocks, suggesting that improving soil structure enhances the protection of POM against decomposition. Similar findings were obtained when comparing different cropping systems in Iowa: POM-C exhibited changes while total SOC did not [52]. Utilizing data from this study, we also discovered a positive correlation between POM-C concentration and potentially mineralizable nitrogen (Figure 5). These results emphasize the importance of POM in defining soil properties and functions. While MAOM is also significant in certain soil functions, particularly nutrient storage and provision [112], it has received less attention [113].

Some forms of soil organic matter are inherently resistant to decomposition. When plant residues are heated at a high temperature in the absence of oxygen (i.e., pyrolyzed), some of the C present is not combusted but is condensed into chemical structures that resist further decomposition. The solid-phase product of pyrolysis is called char or biochar and can be produced at backyard to industrial scales to create a long-lasting soil carbon amendment [114–116]. In addition to removing C from the atmosphere for decadal to centuries-long periods, biochar soil amendments have been shown to enhance beneficial soil properties like cation and anion exchange capacities and water-holding capacity in soils (e.g., [117–120]).

Some soils already have measurable concentrations of such persistent C. These are hypothesized to be the legacy of naturally occurring fires, of fires created by Indigenous hunters to drive game, of localized and intentional soil amendments by Indigenous farmers before modern agriculture, or of crop residue burning to facilitate the subsequent planting of crops. Studies of Iowa soils have reported that 29–36 percent of the C in surface horizons was composed of such "recalcitrant" forms [121]. This fraction represents SOC that is likely to remain stable in the soil for long periods under many variations of soil management practices—as long as the soil is not eroded. The concentration of such recalcitrant C, which

is not quantitatively known for most soils, also has a significant impact on the uncertainty of predictions of soil C stocks that are made using dynamic models [122].

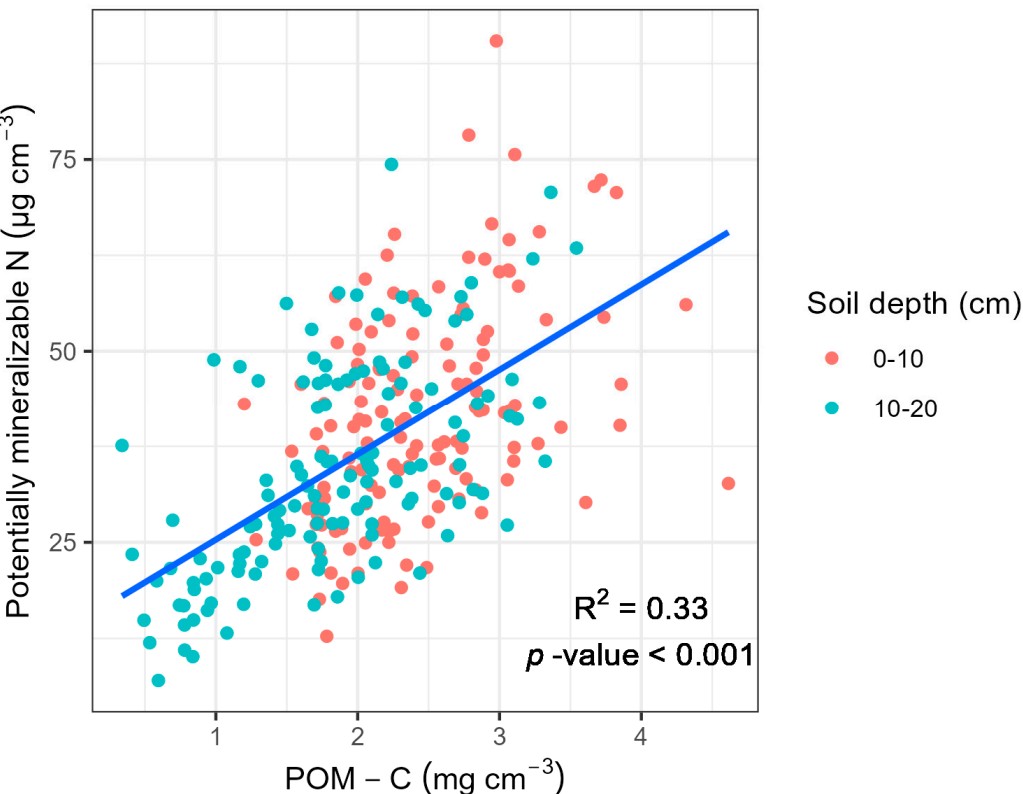

**Figure 5.** Particulate organic matter carbon (POM-C) relationships with potentially mineralizable nitrogen from Marsden Agricultural Diversification Experiment mentioned in previous section (data from [52]).

There are other soil C pools, usually either measured with an extraction or laboratory incubation, that are implicitly included in MAOM-C. This includes microbial biomass C, or C that is contained in living microorganisms, typically measured using chloroform fumigation-extraction or substrate-induced respiration [123]. Microorganisms in soil constantly reproduce and die at rates that depend on temperature and the availability of water and nutrients. While it is difficult to quantitatively differentiate the carbon associated with living versus dead microorganisms, recent research has estimated that in many cultivated and grassland soils the total necromass can account for as much as 50% of the total organic C in the soil [124,125]. Much bacterial biomass and necromass occurs in the clay size fraction, but fungal biomass also extends to the silt size fraction. Microbial biomass is dominated by the cell walls of the organisms, and those walls are composed of polymers of amino sugars. The breakdown of these polymers provides low-molecular-mass organic compounds that other microorganisms and plants use for energy, respiring the carbon as $CO_2$. In addition, the N in those polymers serves as an important source of mineralized inorganic N that both plants and soil organisms require. The importance of MAOM components like microbial biomass and necromass in N mineralization has recently been reported [126].

Also, the low-molecular-weight organic C that can be extracted with water (WEOC) has received some recent attention as it is one of three components in the "Haney Test" used in the Midwest US for soil health assessments [127,128]. Yet another way to fractionate SOC is to use soil biota themselves to produce $CO_2$ in a laboratory incubation—this method of re-wetting dry soils produces a "biologically active" C pool that is typically < 10% of SOC and has been used to predict soil nitrogen supply to crops [129–131]. These methods all

show promise as either proxies for measuring some soil function, like the ability to supply nitrogen, or as indicators of soil health [132,133].

POM has sometimes been treated as a "leading indicator" of long-term changes in organic matter concentrations that result from a change in management practice such as cover crops or reduced tillage. While changes in POM concentrations are more likely to occur over periods of a few years than are changes in total organic matter concentrations, only a minor fraction of POM C will remain unmineralized in most cultivated soils [134]. Thus, predicting the impacts of a management practice on long-term carbon storage by its impact on POM concentrations is problematic. It is also important to note that the most widely used dynamic process models to predict gains or losses of soil organic C are not based directly on POM and MAOM measurements. Models that are based on the CENTURY/DayCent framework (e.g., COMET-Farm, https://comet-farm.com/Home, accessed 17 August 2023) make assumptions about the litter pool (which includes particulates larger than POM), the microbial biomass pool, and the total soil organic matter pool. More recent academic models that include measured POM and MAOM values as input variables have been developed [135–137], but their ability to predict outcomes of soil management practices on soil C has not been tested.

### 4.2. Spatial and Temporal Variation in SOC

Many factors complicate the accounting of SOC stocks, including spatial and temporal variation. Soil samples are intended to represent a soil landscape, which has spatial connectivity. This condition presents challenges and opportunities. For example, lateral movement challenges assumptions made when comparing repeated location samples. Conversely, the fact that SOC stock is a spatially continuous variable presents opportunities to utilize geographic concepts to reduce uncertainty in SOC estimation for a given area.

The spatial modelling of soil carbon stocks uses auxiliary information such as spatial autocorrelation and remotely sensed covariates to improve the representation of the entity [138,139]. Using this additional information can lessen the uncertainty in assessment of carbon stocks compared to using sampling alone [140]. Similar to the question of how much precision is needed from lab measurements for detecting change, spatial modeling methods also contribute to the uncertainty in calculation of SOC stocks. To illustrate the range of outcomes possible by using different methods to assess the amount of SOC in a field, a variety of spatial modeling approaches were applied to a set of 143 soil samples from a 16-ha field in northern Iowa (Figure 6). Differences among the maps illustrate the potential variability in stock estimation resulting from differences in spatial modeling method and sampling density. The impact of the spatial modeling method is expected to increase for areas with higher variability of SOC stock due to the higher magnitude of spatial differences as the method interacts with the spatial pattern. The combination of field data with remotely sensed data dramatically improves the spatial resolution of prediction. These more highly resolved datasets can inform the precision management of SOC, particularly with regard to the strategic spatial integration of perennial crops or conservation cover [141]. As previously discussed, some portions of a field are likely to have a greater potential to accrue SOC than others.

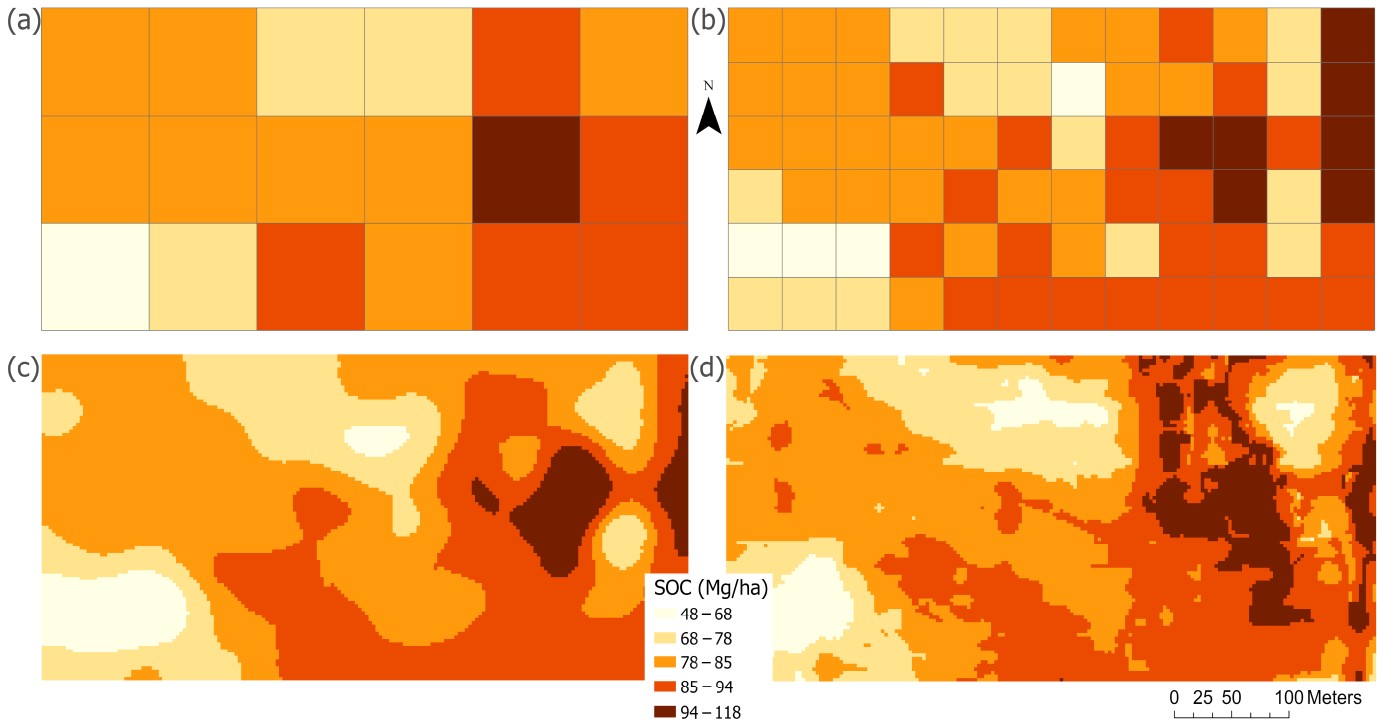

**Figure 6.** SOC stock maps generated from 143 samples (**a**) "composited" to a 0.4-ha (1-ac), (**b**) "composited" to a 0.1-ha (0.25-ac) grid, (**c**) interpolated using ordinary kriging, and (**d**) used to train a machine learning model with Cubist [142] and remotely sensed covariates.

Appropriate choices for soil sampling design depend on the intended use of that data. If they will be used estimate a field mean, then the goals of representing variability and having enough samples for statistical significance will be the most important. In terms of representing variability, the sampling design should be cognizant of equally representing zones versus the spatial weighting of those zones. A grid sampling design is inherently spatially weighted. If a stratified sampling design is applied, such as landscape position or soil map unit, then a strategy for spatially weighting those stratification zones will be needed for estimating the spatially weighted mean. For example, a field with a skewed distribution of SOM could be represented as a normal distribution with strategic sampling.

The number of samples needed also depends on how that data will be used to represent the field and how much uncertainty is tolerable. When the field is modelled by estimating a mean, field variability will often drive the need for a high number of samples to be able to detect significant change [93,143]. Other approaches to model a field can use geographic principles to spatially predict C stocks in unsampled locations. Spatial interpolation methods (e.g., kriging) rely on spatial autocorrelation to fill in values for the field. Spatial regression methods utilize other, more readily measured variables that tend to covary with the target variable to infer values for the complete field. In theory, the inclusion of additional information should reduce uncertainty in the accounting of soil properties in a field. However, the methods for estimating uncertainty differ among these approaches. More work is needed on comparing these methods and determining which are best suited for monitoring soil carbon stocks.

Temporal variation in SOC accrual rates also complicate accounting. As noted earlier in the context of POM and MAOM fractions, some scientists have hypothesized that soils have an intrinsic, "saturation" limit, that is a steady state status beyond which carbon stocks will not increase; in other words, the rate of carbon addition will be equal to the rate of its loss [105,107,144]. The carbon saturation concept is focused not on how much total carbon can accumulate in a soil, i.e., stored carbon, but on how much might be sequestered or stabilized for long periods, which depends mainly on clay-size minerals—if all other factors

are held constant. Scientists promoting the saturation concept reason that the primary control over the amount of carbon that is stabilized and not subject to loss is the mineral surface area that protects organic matter from being broken down by microbial enzymes. This hypothesis has been used to explain why increasing carbon stocks can be fast in some soils and very slow in others. However, there are some soils in Iowa almost entirely composed of organic matter with little mineral material (Histosols). These soils suggest that, under the right conditions, there is no limit to how much carbon can accumulate in some parts of the landscape.

While the protection of organic matter in association with minerals is one way C can be stabilized, some organic matter associated with clay and silt particles remains susceptible to oxidation. The period required to reach stable carbon stocks will vary not only with the quality and quantity of organic matter inputs but also with soil management practices (e.g., tillage, fertilization), changing climate, and vegetation. Soil amendments of organic matter in forms that are already resistant to decomposition, such as pyrolyzed biochar, will increase SOC in a soil beyond its saturation limits defined by texture. Digestate solids that result from the breakdown of organic matter through anaerobic digestion are also very slow to breakdown after land application and could increase SOC beyond typical saturation limits [145]. These soil management strategies require more research, especially through long-term replicated field trials in which several management factors are investigated in combination.

One concern moving forward is that studies of the past effect of management practices on SOC may not be predictive of future carbon storage given climate change. Climate change has likely contributed to increased maize and soybean yields in Iowa in recent decades due to the timing of humidity, rainfall, and heat [146]. Moreover, Iowa has had the good fortune of not experiencing the same level of warming to date as the global trend (aka, "the warming hole"). This is due in part to a buffering effect on peak summer and early fall temperatures as a result of crop evapotranspiration of spring and early summer rainfall and the high water holding capacity of Iowa's soils [147]. This advantage may be diminishing, as increasing spring rainfall often delays planting and increases soil erosion, and rising humidity increases the growth of molds and fungi [146]. Under a business-as-usual scenario in terms of global emissions, Iowa is likely to experience severe droughts in between high rainfall years from the present to ~2050, and subsequently, high temperatures, heat stress, and reduced growing season rainfall will tend to significantly constrain crop yields [146,148,149]. These changes will likely adversely impact the ability of Iowa soils to sequester carbon. Integration of the DAYCENT model with various Intergovernmental Panel on Climate Change (IPCC) climate models for central Iowa suggests declines in SOC in the latter part of 21st century, largely due to harvest losses, despite increases in net primary productivity [150].

The SOC benefits of converting areas of row crops to perennial grasses are likely to continue, even if slowly. However, profitable uses of perennial grasses are desperately needed to combine this clear SOC benefit with economic viability [151]. There is some nascent research on using perennial grain crops like Kernza® to concomitantly obtain the combined outcomes of grain production and SOC benefits [152,152]. Further research is underway on using perennial, native grasses as plant biomass feedstock for anaerobic digestion [151]. A clear path for recovering SOC, and sustainable agriculture, will likely require cropping systems that maintain continuous living cover, whether through integrating winter crops or through perennial plants: grain crops, forage crops, biomass crops, or restored native grassland species.

*4.3. Other Considerations*

Another major factor that may affect long-term SOC storage in Iowa soils is artificial drainage. Tile drainage has been used in Iowa since the 19th century to lower water tables and to remove excess soil water to allow earlier planting and to reduce nitrogen loss by denitrification during the growing season. Today, close to 60% of Iowa's cropland

is tiled [58,153,154]. Artificial drainage may also have an impact on the storage of soil carbon because it increases aeration and soil temperatures [155], potentially leading to the increased microbial oxidization of organic carbon [154,156,157].

Cropland soils in low-lying topographic positions tend to have higher soil carbon stocks [158] (e.g., the poorly-drained Webster soils vs. well to moderately well-drained Clarion soils in the same landscape (Figure 1)) because organic-matter-rich eroded sediments accumulated there or because decomposition rates of plant biomass were slow before agricultural exploitation. Subsurface drainage systems can release soil carbon from soils in these positions as $CO_2$ by lowering the water table, aerating the soil, and promoting decomposition of organic matter. Carbon markets have started to recognize the impacts: adding or improving drainage after the start of a project is not allowed in common soil carbon protocols (e.g., [159]). While subsurface drainage may promote soil carbon loss as $CO_2$, modeling studies suggest that artificial drainage could, in some years, reduce direct nitrous oxide ($N_2O$) emissions from maize cropping systems and may thus lead to lower net cumulative GHG emissions [154]. This is important to consider alongside carbon programs because one ton of $N_2O$ traps 265 times more heat in the atmosphere than one ton of $CO_2$ over a 100-year period [160].

The emission of $N_2O$ is most likely when soils are nearly but not completely saturated with water and the level of oxygen in the soil is low. Long-term measurements from typical maize/soybean rotations on poorly and somewhat poorly drained soils in central Iowa indicate that $N_2O$ emissions from soil can be a significant contribution to atmospheric greenhouse gases [161,162], and they can release more $N_2O$ than well-drained soils in other parts of the Midwest where previous research has been conducted [163]. Water-filled pores in an otherwise unsaturated soil can host microorganisms capable of biochemically reducing oxidized forms of nitrogen, like nitrate, to $N_2O$ gas. Other studies suggest that $N_2O$ emissions associated with practices like no-till, cover crops, and rotations are not likely to offset the lowered warming potential that results by $CO_2$ removal using those practices (e.g., [164]). More field research is needed to accurately predict $N_2O$ emissions from soils under a greater variety of management regimes. The most important management practices to limit $N_2O$ emissions from Iowa soils are already well known, however, and include (1) timing application of nitrogen fertilizers to crop growth periods when nitrogen is most needed, and (2) effective and timely drainage of somewhat poorly and poorly drained soils during spring snow melt and during the growing season.

Given the cost of data field collection, SOC cannot be sampled and understood at spatial and temporal resolutions sufficient to fully inform each management decision—predictions are necessary. Simulation models play a crucial role by providing a mathematical framework to represent our understanding of SOC dynamics and make predictions under various environmental conditions. Voluntary carbon credit programs rely on such simulation models, and sometimes also on sophisticated statistical analysis, to defensibly align sampling and analysis costs with carbon market values [8,9]. Agroecosystem models now are reasonably adept at prediction of maximum and minimum values from practice changes, but providing predictions of variability at field scale is less robust [9]. While soil measurements from Iowa are more extensive than many regions of the world, data to support model development and use are still limited in several respects. They are limited in both spatial resolution, including depths below 30 cm, and temporal resolution. They are particularly scant regarding the impacts of artificial drainage and newly emerging practices including multispecies cover crops and soil amendment with biochar or digestate [9].

Additional research is needed to improve SOC models. There is further need for models to provide more holistic predictions, integrating other crucial factors, including robust estimates of changes in GHG emissions, crop yield, etc., along with changes in SOC stocks [9]. Collaboration among modelers with researchers who generate empirical data is particularly essential, as the empirical data obtained from experiments is not always the most suitable for these models, and vice versa [165]. For instance, traditional SOM simulation models typically divide SOC into different conceptual pools (usually 3–5 pools,

depending on the model) to represent SOC heterogeneity (e.g., [166,167]); however, most of these pools do not correspond to the readily measurable fractions. This limitation significantly impacts the model's application, calibration, and validation since values for the input variables are assumed from limited and legacy empirical data rather than from contemporary measurements. A new generation of SOC models based on measurable fractions [135–137] and aimed at addressing this issue, represents an example of improved communication between modelers and experimental scientists.

## 5. Conclusions

There is growing interest in improving the measurement and management of SOC in agroecosystems due to its linkages to soil health and crop productivity, agroecosystem resilience under climate change, developing voluntary C markets, and Scope 3 emissions documentation. In assessing results from long-term studies on the effect of agricultural management on SOC stocks in Iowa—a relatively well studied part of the world—we found increases in SOC of 0.25–0.43 Mg C ha$^{-1}$ yr$^{-1}$ with adoption of strip-tillage and no-tillage compared to losses of 0.24 to 0.46 Mg C ha$^{-1}$ yr$^{-1}$ with more intensive after tillage methods; highly variable and inconsistent effects of diversifying crop rotations with extended rotations on SOC; and increases in SOC of 0.21–0.74 Mg C ha$^{-1}$ yr$^{-1}$ with conversion of cropland to perennial grassland. Results further indicate that SOC is highly variable spatially and responds slowly to management changes, and that changes under some management practices are difficult to detect with statistical certainty. While these general conclusions are not scientifically surprising, they are of increasing importance as society grapples with managing atmospheric $CO_2$ levels alongside feeding eight billion people and delivering other important ecosystem services. Given this constellation of factors, we suggest improved prediction of changes in SOC requires the development and use of new methods that can identify and disentangle multiple sources of variability; looking beyond total SOC and toward systematic collection of data on more responsive and functionally relevant fractions (including but not limited to POM-C and MAOM-C); monitoring the effects on SOC of multiple conservation practices used in combination in long-term studies; whole-profile SOC monitoring; and deeper collaboration between field soil scientists and modelers, which can help design the long-term studies needed to fill knowledge gaps.

**Author Contributions:** Conceptualization, J.J., M.M., B.A.M., M.T. and L.A.S.; Data curation, M.M., B.A.M. and M.T.; Writing—original draft, J.J., M.M., B.A.M., M.T., S.V. and L.A.S.; Writing—review & editing, J.J., M.M., B.A.M., M.T., S.V. and L.A.S. All authors have read and agreed to the published version of the manuscript.

**Funding:** This research was funded by USDA-NIFA, grant 2020-68012-31824; the Walton Family Foundation, grant 00104781; USDA National Research Initiative Grant Number 2005-35107-16128; and USDA-NIFA, Hatch project No. IOW05614, 1020365.

**Data Availability Statement:** No new data were created or analyzed in this study. Data sharing is not applicable to this article.

**Conflicts of Interest:** The authors declare no conflict of interest.

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
