# Peer review of "Carbon Storage in Cropland Soils: Insights from Iowa, United States"

_land, doi:10.3390/land12081630_

Round 1

Reviewer 1 Report

In my opinion, under the title and authors list, only correspondence author e-mail must be included, but this is the competence of the Editors of the journal.There are some grammatical errors, for example, “Coeffiient of Variation” (fig.4, y axes). The meteorological conditions of the areas described should also be mentioned, at least to a minimum. In the Conclusions part, there is a lack of generalization of what benefits the collected material and data bring, what specific Conclusions are drawn. It just postpones everything for the future, what should be done.... In my opinion, this reduces the value of work. Abstract is arranged more specifically, showing the regularities found.

In fact, it is very difficult for me to assess the level of this type of article. It is much simpler to evaluate an article prepared by the authors of data from specific experiments. Still, the article shows some picture of the carbon status in Iowa, and therefore could be of interest to some researchers and readers.

Reviewer 2 Report

- units Mg C /ha/yr - probably is better to use kg C/ha/yr

- rainfall amount is better to use in mm (not in cm)

- table 1 - it would be better to state all textural fraction composition, not only silt, especially the content of clay according to USDA

- Mollic Thickness - is it a depth of mollic A horizon? It would be suitable to add next important data characteristic for mollic horizons

- table 2 - the range of sampling depth is very wide, various horizons can be mixed and objectivity of obtained results  can decrease

I recommend submitted paper after minor revision as a review.

Reviewer 3 Report

The manuscript is well-written and grounded in its approach. It is interesting research and of great interest to Land Journal readers.  The manuscript is very long with more than 151 Reference list.   In its current version, the manuscript has a few minor aspects that need to be improved, which were pointed out in the pdf file throughout the text :

Page 5/ Line 139  - Pg or Mg ?

Carbon Storage in Cropland Soils: Insights from Iowa

The shrift of text is different in some places (Calibri ; Palatino Linotype, etc.)

Page 10 line 114 (Ibrahim et al. 2018), and iii) - different citation?

Page 14 line 114 - data from Lazicki et al. 2016 different citation?

Page 14 line 105 - Utilizing data from this study, we also discovered a positive correlation between POM-C concentration and potentially mineralizable nitrogen, and a negative correlation between

Page 16 Line 175 -  Soil organic carbon stock maps generated from 143 samples a) 'composited' to a one-acre  grid, b) 'composited’ to a 1/4-acre grid, c) interpolated using ordinary kriging, and d) used to train  a machine learning model (Cubist) with remotely sensed covariates – Please give citation .

Page 18 Line 286 - wholistic predictions or holistic ? 

 Correlation is too low in Figure 5  -  POM-C and bulk density. 

The conclusion is too common . Please give more detailed information, as in the abstract.

In References

Line 334 - Baum, C.M.; Buck, H.J.; Burns, W.; Butnar, I.; Delerce, S.; Forsell, N.; Fridahl, M.; Haunschild, R.; Höglund, R.Honegger, M.; et al. The State of Carbon Dioxide Removal  - Add  year,  add source (journal, book, issue ……….) !

Line 250 – again, Please add year and issue .

Line 585 – Add  year . 
